# Construction of Three-Dimensional Network Structure in Polyethylene-EPDM-Based Phase Change Materials by Carbon Nanotube with Enhanced Thermal Conductivity, Mechanical Property and Photo-Thermal Conversion Performance

**DOI:** 10.3390/polym14112285

**Published:** 2022-06-04

**Authors:** Yunbing He, Yanfeng Chen, Cuiyin Liu, Lisha Huang, Chuyu Huang, Junhua Lu, Hong Huang

**Affiliations:** 1Guangdong Provincial Key Laboratory of Distributed Energy Systems, School of Chemical Engineering and Energy Technology, Dongguan University of Technology, Dongguan 523808, China; heyb@dgut.edu.cn (Y.H.); lisa_huang.dg@foxmail.com (L.H.); hcy070514@163.com (C.H.); ljhuas@163.com (J.L.); 2School of Chemistry and Chemical Engineering, South China University of Technology, Guangzhou 510640, China; cehhuang@scut.edu.cn; 3School of Materials Science and Hydrogen Energy, Foshan University, Foshan 528000, China; liu_cuiyin@163.com

**Keywords:** carbon nanotube, three-dimensional network, thermal pathway, mechanical property, photo-thermal performance

## Abstract

High thermal conductivity and good mechanical properties are significant for photo-thermal conversion in solar energy utilization. In this work, we constructed a three-dimensional network structure in polyethylene (PE) and ethylene-propylene-diene monomer (EPDM)-based phase change composites by mixing with a carbon nanotube (CNT). Two-dimensional flake expanded graphite in PE-EPDM-based phase change materials and one-dimensional CNT were well mixed to build dense three-dimensional thermal pathways. We show that CNT (5.40%wt)-PE-EPDM phase change composites deliver excellent thermal conductivity (3.11 W m^−1^ K^−1^) and mechanical properties, with tensile and bending strength of 10.19 and 21.48 MPa. The melting and freezing temperature of the optimized phase change composites are measured to be 64.5 and 64.2 °C and the melting and freezing latent enthalpy are measured to be 130.3 and 130.5 J g^−1^. It is found that the composite phase change material with high thermal conductivity is conducive to the rapid storage of solar energy, so as to improve the efficiency of heat collection.

## 1. Introduction

With the rapidly growing demand of renewable energy, efficient heat storage and energy conversion are becoming principal themes in today’s energy society [1,2,3]. As solar energy is an inexhaustible and worldwide distributed renewable resource, solar thermal utilizations have been broadly applied in energy storage in recent decades [4,5]. Over the past few decades, phase change materials (PCMs) have been widely used in solar energy storage due to their reversible heat storage and exothermic properties in the course of the phase change process [6,7]. PCM can store immense heat by gathering the sunlight directly in the temperature range of phase transition, hence it has been used as an excellent heat storage medium [8].

The original PCMs include paraffin wax, fatty acid, polyol, and so on. However, because of the inherent defects of original PCM, such as leakage, low thermal conduction coefficient and weak light absorption, the application of PCMs is subject to considerable restrictions. To solve the leakage of original PCMs, porous supported materials were used to absorb the original PCMs by capillary force and surface tension when heating to liquid [9,10,11,12], or original PCMs as core material and polymers as shell materials were used to prepare phase change microcapsules [13,14,15]. The porous supported material-based PCMs and phase change microcapsules could both effectively prevent the leakage of original PCMs through heating. When the porous carbon materials such as expanded graphite (EG) [16,17], graphene foam [18,19], carbon nanotube sponge [20], and ordered mesoporous carbon (CMK-3) [12] were used as the supported materials, the thermal conduction coefficient of the as-prepared PCMs could also be largely increased. However, the PCMs prepared by the ways mentioned above have relatively poor mechanical properties, especially when the temperature is higher than the phase change temperature range. For the continuous utilization of solar thermal energy, a good mechanical property is required for PCMs to resist the strong scouring of fluids during the heat exchange procedure between the material and fluids. To boost the mechanical properties of PCMs, polymer materials were introduced as the supporting skeleton to combine the original PCMs by using polymer processing technology. The as-prepared polymer-composited PCMs would present much larger tensile and bending strength than that of porous material-based PCMs. Among various polymer materials, the ones with good compatibility with paraffin were selected as skeleton materials to ensure no leakage appeared in the polymer-based phase change materials, such as polyethylene (PE) [21], styrene-butadiene styrene block copolymer [22], and olefin block copolymers [23].

In our previous work [24,25], polyethylene (PE) and ethylene-propylene diene copolymer (EPDM) were used to prepare PE-EPDM/EG phase change materials with enhanced mechanical properties. The thermal conductivity was improved by the addition of a small amount of EG, which helped to avert the leakage of paraffin. Furthermore, the effect of PE and EPDM on the compatibility of EG/paraffin phase change materials was studied by analyzing the microstructure, crystal structure and functional groups of the composite. The as-prepared PE-EPDM/EG phase change composite was applied to thermal insulation and photo-thermal conversion of building roofs. By virtue of the improved thermal conductivity, the phase change composite delivers accelerated rates of photo response and heat storage and release. In the PE-EPDM/EG phase change composites, on one hand, EG played the role of adsorbing paraffin, on the other hand, EG was used as a thermal conducting filler for providing a large framework to build the thermal conduction pathway. However, for the characteristics of the two-dimensional flake structure of EG, the thermal conduction pathways between EG flake layers were blocked by paraffin with low thermal conductivity. Carbon nanotubes (CNTs) have superior thermal conductivity, delivering high values of the thermal conductivity coefficient of 6000 W m^−1^ K^−1^ for a single-walled carbon nanotube (SWNT) and 3000 W m^−1^ K^−1^ for a multi-walled carbon nanotube (MWNT) [26]. In comparison to metal or metal oxide materials, CNTs can build a better thermal pathway with a small number of additions.

In this work, we introduced CNTs into the PE-EPDM/EG-based phase change composites to construct a thermal pathway between the isolated area between EG flake layers and paraffin. By adjusting the addition gradient of CNTs, the effect of CNTs on the microstructure, thermal conductivity, mechanical properties and photothermal properties of CNT-PE-EPDM-EG/OP70 phase change materials were investigated. By adding CNTs, the thermal conductivity, mechanical properties and photothermal conversion performance were greatly improved, at the expense of a small loss of enthalpy (5.40%). This work provides a facile approach to preparing polymer-based phase change materials for the efficient utilization of solar thermal energy.

## 2. Materials and Methods

### 2.1. Materials

A multi-walled carbon nanotube (CNT) was obtained from Nanjing Pioneer Nanotechnology Co., Ltd., (Nanjing, China). Technical grade paraffin (OP70, T_m_ of about 65 ℃) was purchased from Shanghai Joule Wax Co., Ltd., (Shanghai, China). Expanded graphite (EG) was obtained from Qingdao Herita graphite Product Co., Ltd., (Qingdao, China). The high-density polyethylene (HDPE, DMDA8920, molecular weight: 40,000–300,000, 0.954 g cm^−3^) was obtained from China National Petroleum Co., Ltd., Beijing, PR. China. The Ethylene propylene rubber (EPDM, 3745P, ethylene content: 70.0%wt, propylene content: 29.5%wt, ethylene-norbornene content: 0.5%wt, 0.88 g cm^−3^) as a random copolymer was purchased from Dow Chemical Company. All chemicals were used as received without further purification.

### 2.2. Preparations of CNT-Polyethylene-EPDM-Based Phase Change Materials

The OP70/EG sample was prepared from EG absorbing OP70 with a mass ratio of 1:9. Then, 70 g of the obtained OP70/EG, 5 g of ethylene propylene rubber and 30 g of high-density polyethylene were mixed through open milling procedure (XH-401, Dongguan Xihua Co., Ltd., Dongguan, China) [24,25]. To boost the thermal conductivity and the mechanical property, 1, 2, 4, 6, 8 g of CNT were added, respectively, following sequential blending. The mass ratio and mass percentage of the as-prepared samples are present in Table 1 and Table 2. The as-prepared phase change composites CNT-PE-EPDM-EG/OP70 were fabricated into blocks by hot-press (XH-406B, Dongguan Xihua Co., Ltd., Dongguan, China).

### 2.3. Characterization of PCMs

The thermal conductivities of the as-prepared CNT-PE-EPDM-EG/OP70 phase change materials were evaluated through a thermal constant analyzer (Hot Disk TPS 2500S, Hot Disk AB, Göteborg, Sweden). An Electromechanical Testing Machine (104B, WANCE Testing Machine Co., Ltd., Shenzhen, China) was used to analyze the mechanical properties of the as-prepared CNT-PE-EPDM-EG/OP70 phase change materials, including the tensile and bending strength. The tensile tests are based on the ISO standard 527-2012, while ISO standard 178-2010 is used for the three-point bending tests. The morphologies of the as-prepared CNT-PE-EPDM-EG/OP70 phase change materials were observed by Field emission scanning electron microscopy (FE-SEM, Hitachi SU8220, 5 kV, Japan). Attenuated total reflection Fourier transform infrared (ATR FT-IR) spectra of the as-prepared phase change materials were collected by a Bruker Vector 33 spectrometer. Raman spectra of CNT/PE-EPDM/EG-based phase change composites were measured by using LabRAM ARAMIS Raman spectrometer (LabRAM HORIBA Jobin Yvon, Edison, NJ, USA) with a HeNe laser as an excitation source. The phase change temperature and latent heat of the CNT/PE-EPDM/EG-based phase change composites were tested using a differential scanning calorimeter (Q20, TA-Instrument, New Castle, DE, USA) at a heating rate of 10 °C min^−1^ in the atmosphere of nitrogen with a flow rate of 50 mL min^−1^.

### 2.4. Photo-Thermal Conversion Performance of CNT/PE-EPDM/EG-Based PCMs

The photo-thermal conversion performance of the as-prepared CNT/PE-EPDM/EG-based PCMs was conducted photothermal conversion test system, in which solar simulator (Microsolar300, Beijing Perfectlight Technology Co., Ltd., China), receiver, thermocouples and data acquisition systems (Agilent 34970A) were contained. A solar receiver was used to deposit the CNT/PE-EPDM/EG-based PCM samples, which is located under the solar simulator with the distance of 10 cm. The temperature change in the CNT/PE-EPDM/EG-based PCM samples was recorded by the thermocouples. A heat storage process of PCMs was carried out with the light turned on, following a heat release process after the lights turned off. An irradiatometer (ST-80C, Photoelectric Instrument Factory of Beijing Normal University, Beijing, China) was used to measure the irradiation intensity of solar simulator. The receiver efficiency is calculated according to Equation (1).
(1)$$\eta =\frac{Q_{T}}{G_{s}At}\times 100\%$$
where the *η* is the receiver efficiency, *G_s_* is the irradiance of solar radiation, *A* is the irradiation area, *t* is the irradiation time.

## 3. Results and Discussion

### 3.1. Thermal Conductivity Analysis of the as-Prepared CNT/PE-EPDM/EG-Based PCMs

The thermal conductivities of the as-prepared CNT/PE-EPDM/EG-based PCMs are measured and shown in Figure 1. It is found that the PCMs exhibit gradually increased thermal conductivity with the increasing amounts of CNT. The thermal conductivity coefficient of PE-EPDM/EG-based PCM without the addition of CNT is 1.99 W m^−1^ K^−1^, and it increases to 2.14 W m^−1^ K^−1^ when 0.94 wt% CNT was added. It is noteworthy that the thermal conductivity increases almost linearly while the mass fraction of CNT increases from 0 to 5.4%. However, the thermal conductivity coefficient increases from 3.11 to 3.20 W m^−1^ K^−1^ in the case of increasing the mass fraction of CNT from 5.40 to 7.08%, indicating a reduced degree of growth. This result suggests that the thermal pathway of CNT/PE-EPDM/EG-based PCMs were mainly constructed with the mass fraction of 5.40% and the effect of further additional CNT on the improvement of thermal conductivity was gradually weakened.

### 3.2. Morphology and Microstructure of the as-Prepared CNT/PE-EPDM/EG-Based PCMs

In our previous work [24,25], it is indicated that the morphology of PCMs is affected with the compatibility and compactness of PE and EPDM mixture. The EPDM has good compatibility with EG/paraffin, and PE has good compactness on the surface. Combined with the PE and EPDM, the surface of PCMs tend to coarse and compact; the PCMs might combine the compatibility of EPDM and the mechanical properties of PE. In order to explore the mechanism on the improvement in thermal conductivity for the as-prepared CNT-PE-EPDM/EG-based PCMs, the morphologies of the samples were observed by SEM (Figure 2). As shown C1 in Figure 2, it can be observed that EG, PE and EPDM intertwine with each other, forming a network architecture. With the addition of 0.94 wt% CNT, the stripped carbon tubes are further wound on the EG-PE-EPDM network structure (C2 in Figure 2). Compared to the sample without CNT, the sample with 0.94% mass fraction of CNT has a more compact winding structure. As shown from C2–C6 in Figure 2, with the CNT mass fraction increase from 0.94 to 7.08%, the wound network structure becomes more and more dense, while the sheet structure of EG disappears gradually. This is due to the wrapping of CNT on the surface on the sheet structure of EG/OP70. The increasing amount of CNT can make the surface of EG/OP70 form a denser envelope, which favors the construction of more phonon transport channels, and the formation of more thermal pathways. Building such a dense structure can not only prevent the leak of OP70 from phase change materials, but also improve thermal conductivity, together with the capture and divergency of photons so as to accelerate the heat transfer rate and the photothermal conversion performance.

The possible enhancement mechanism of the thermal conduction of the CNT-PE-EPDM/EG matrix composite phase change materials prepared is shown in Figure 3. A one-dimensional carbon nanotube with high thermal conductivity was added to flake expanded graphite composites to form a network wound structure, then a three-dimensional network architecture was formed form the dimensional expanded graphite phase change composites and the one-dimensional carbon nanotube. Thus, enhanced thermal conductivity was obtained for the phase change composite.

### 3.3. Mechanical Property of the as-Prepared CNT/PE-EPDM/EG-Based PCMs

The tensile and bending strength of the as-prepared CNT/PE-EPDM/EG-based PCMs were measured. As presented in Figure 4A, the tensile strength of the PCMs increases from 8.39 MPa (sample C1) to 10.19 MPa (sample C5) in the case that the mass fraction of CNT increases from 0.94 to 5.40%. However, the tensile strength of sample C6 decreases to 9.11 MPa, while the mass fraction of CNT increased to 7.08%. Similarly, as shown in Figure 4B, the bending strength of the PCMs increases from 14.67 MPa (sample C1) to 21.48 MPa (sample C5), but the bending strength of sample C6 decreases to 13.65 MPa, while the mass fraction of CNT increased to 7.08%. This indicated that a small amount of CNT could promote the structure of CNT/PE-EPDM/EG-based PCMs compact and enhance the tensile and bending strength. However, when the addition amount exceeds the critical value, the continuous structure of polyethylene-EPDM/EG-based PCMs is destroyed, causing a sharp drop in mechanical strength.

### 3.4. FT-IR and Raman Analysis of the as-Prepared CNT/PE-EPDM/EG-Based PCMs

The ATR FT-IR spectrograms of PE-EPDM/EG-based PCM sample (C1) and CNT/PE-EPDM/EG-based phase change composite (C5) are shown in Figure 5. For the PE-EPDM/EG-based PCM sample (C1), the bands located at 722, 1018 and 1103 cm^−1^ can be assigned to the vibration of C-H and rocking vibration of =C-H and C-H, respectively, which are ascribed to the groups from alkanes or alkenes of paraffin. The band located at 942 cm^−1^ can be assigned to the bending vibration of O-H, which corresponds to carboxyl of expanded graphite [24]. The bands at 1297, 1375, 1411, 1431 and 1464 cm^−1^ can be assigned to the C-H swing vibration modes, C-H vibration modes, C-C stretching vibration modes and C-C heterocycle vibration modes, respectively. The bands at 1700 cm^−1^ are contributed to the C=O stretching vibration modes of carboxyl in expanded graphite. The bands at 2849 and 2916 cm^−1^ are associated with the stretching vibration modes of -CH_2_ and -CH_3_ groups in paraffin. With the CNT mass fraction of 5.40%, the intensity of the infrared characteristic absorption peaks of paraffin in CNT/PE-EPDM/EG-based phase change composite (C5) are greatly weakened. This indicates that the addition of CNT can effectively improve the encapsulation of paraffin, which is beneficial for preventing the leakage of paraffin in CNT/PE-EPDM/EG-based PCMs.

The Raman spectra of PE-EPDM/EG-based PCM sample (C1) and CNT/PE-EPDM/EG-based phase change composite (C5) are illustrated in Figure 6. For PE-EPDM/EG-based PCM sample (C1), the bands at 1058, 1123, 1283 and 1431 cm^−1^ are attributed to asymmetric stretching vibration between carbon atoms, symmetric stretching vibration between carbon atoms, carbon and hydrogen atoms swing between vibration and asymmetric bending vibration between carbon and hydrogen atoms, respectively, which are mainly produced by polyethylene molecules in PE-EPDM/EG-based PCM. Absorption bands at 2719, 2843 and 2896 cm^−1^ contribute to the vibration modes of -CH_2_ groups in paraffin. The emergence of a D band located at 1350 cm^−1^ can be ascribed to the defects in EG or CNT. The G band located at 1580 cm^−1^ is assigned to the E_2g_ vibration mode of sp2 carbon, which reflects the graphitization degree of EG or CNTs.

### 3.5. Melting and Freezing Behavior of CNT/PE-EPDM/EG Phase Change Material

The DSC curves and melting-freezing properties of the as-prepared PE-EPDM/EG-based PCM samples are compared in Figure 7 and Table 3. The melting and freezing temperatures of PE-EPDM/EG-based PCM are 63.8 and 63.9 ℃, corresponding to the latent enthalpies of 138.9 and 138.4 J g^−1^, respectively. With the increasing amount of CNT from 0.94 to 7.08wt%, the melting latent enthalpy of CNT/PE-EPDM/EG-based phase change materials decrease from 137.6 to 129.1 J g^−1^. The enthalpy of CNT/PE-EPDM/EG-based phase change material is reduced at a minor value of about 7.9 J g^−1^ and its thermal conductivity is improved from 1.99 to 3.11 W m^−1^ K^−1^ with a small amount of CNT addition (C1 to C5). It is worth noting that highly similar curves were collected for all the PCMs samples. Low heat transfer lag during the DSC test was observed due to their high thermal conductivities.

### 3.6. Reversible Property of CNT/PE-EPDM/EG Phase Change Materials

In order to evaluate the cyclic thermal stability of CNT/PE-EPDM/EG phase change material, a heating–cooling test with 200 cycles in the temperature range 10–80 ℃ was carried out. The reversible stability of CNT/PE-EPDM/EG phase change materials was evaluated by DSC and SEM. The morphology of CNT/PE-EPDM/EG phase change material after 200 heating–cooling cycles was observed and shown in Figure 8. The remaining similar structure suggests that the as-prepared CNT/PE-EPDM/EG phase change material has good cycling stability. The melting and freezing behavior of the PE-EPDM/EG phase change composite after 200 cycles of heating–cooling treatment was tested to further verify the reversible stability (Figure 9). The as-prepared CNT/PE-EPDM/EG phase change material exhibited coincident latent heat values before the heating–cooling test and after the cycling treatment, including 130.3 and 129.9 J g^−1^ for melting and 130.5 and 130.1 J g^−1^ for solidification. Both highly reversible curves indicate the good thermal stability of the as-prepared CNT/PE-EPDM/EG phase change material.

### 3.7. Photo-Thermal Conversion Performance of CNT/PE-EPDM/EG Phase Change Materials

The photo-thermal conversion performance and receiver efficiency of the as-prepared CNT-PE-EPDM/EG composite PCMs are shown in Figure 10. It can be found that the temperature plateau in the range 65~70 ℃ appears in the both heating and cooling process, showing the typical characteristics of the temperature-time curve of phase change materials. The time for sample C1 to increase from room temperature to 98.3 ℃ is 274 s, while the times for sample C2 to C6 are 237, 211, 195, 166 and 163 s, respectively. From sample C1 to C6, the temperature rise time is reduced by 40%. These results indicate that the temperature growth rate is positively correlated with the increasing thermal conductivity. The receiver efficiency is calculated according to Equation (1). The calculated receiver efficiencies are shown in Figure 10B. It is shown that the receiver efficiency of all samples presents a downward trend with the raise in temperature. This is mainly because the higher temperature, the more heat flows into the atmosphere. The receiver efficiency of sample C1 decreases from 70% to 50% as the temperature rises from 30 to 90 ℃. The receiver efficiency from sample C1 to C6 increases, in turn, under the same receiver temperature. The receiver efficiency of sample C5 decreases from 87% to 74% at as the temperature, rising from 30 to 90 ℃. The receiver efficiency of sample C6 reaches 88% at 30 ℃ and over 80% at 90 ℃. These results demonstrate that high thermal conductivity in CNT/PE-EPDM/EG phase change material is beneficial for the rapid storage of photo thermal conversion energy, and is thus conducive to the improvement of the receiver efficiency of phase change material.

## 4. Conclusions

In this work, we constructed a three-dimensional network structure in polyethylene and ethylene-propylene-diene monomer-based phase change materials by adding a carbon nanotube. The as-prepared CNT-PE-EPDM/EG-based phase change material with high thermal conductivity and good mechanical properties exhibits high latent heat and good photo-thermal performance for the effective utilization of solar thermal energy. The optimized mass fraction of CNT in CNT-PE-EPDM/EG phase change composite is 5.40%. The optimized CNT-PE-EPDM/EG-based phase change material has a thermal conductivity of 3.11 W m^−1^ K^−1^ with the constructed dense three-dimensional thermal pathways by CNT addition and has good mechanical properties, with tensile and bending strength of 10.19 and 21.48 MPa. The melting and freezing latent enthalpy of the PCM are estimated to be 130.3 and 130.5 J g^−1^, respectively. The FT-IR and Raman results indicate the addition of CNT can effectively improve the encapsulation of paraffin for preventing the leakage of paraffin in CNT/PE-EPDM/EG-based PCM. The CNT/PE-EPDM/EG-based PCM shows great photo-thermal performance with 60.6% of heating storage time and 1.48 time of receiver efficiency than PE-EPDM/EG-based PCM. The CNT/PE-EPDM/EG-based PCM is expected to be one of the advanced materials for photo-thermal energy storage in solar thermal utilization. This work provides a simple way to prepare a PCM combined with great mechanical properties, high thermal conductivity, large latent heat and good photo-thermal performance. In particular, PCMs with good mechanical properties could withstand the scouring of fluids in the heat exchange process between the PCMs and other fluids.

## Figures and Tables

**Figure 1 polymers-14-02285-f001:**
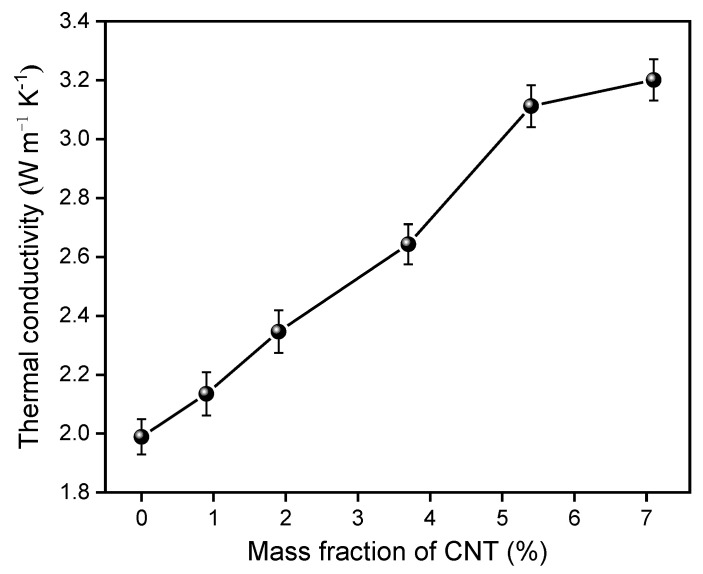
Thermal conductivity of the as-prepared CNT-PE-EPDM/EG-based PCMs.

**Figure 2 polymers-14-02285-f002:**
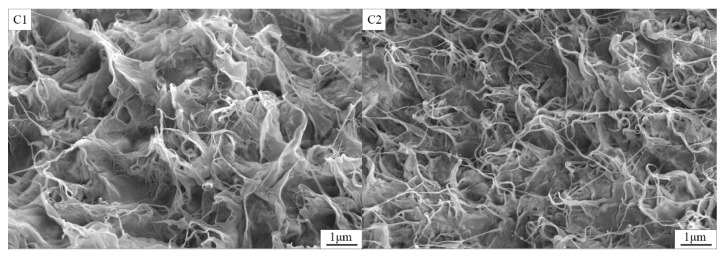
SEM images of CNT/PE-EPDM/EG-based phase change composites (**C1**–**C6**).

**Figure 3 polymers-14-02285-f003:**
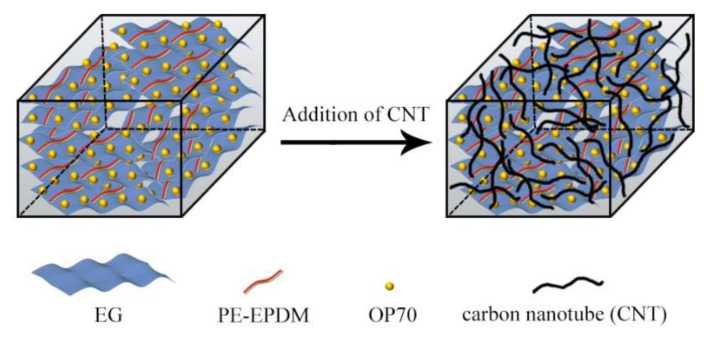
Mechanism of thermal conductivity enhancement for CNT-PE-EPDM/EG-based PCMs.

**Figure 4 polymers-14-02285-f004:**
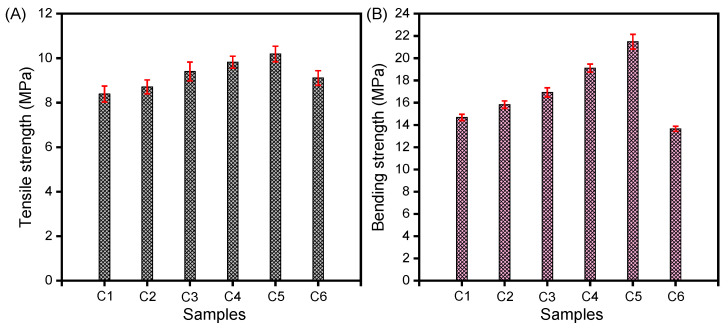
Tensile strength (**A**) and bending strength (**B**) of CNT-PE-EPDM/EG-based PCMs.

**Figure 5 polymers-14-02285-f005:**
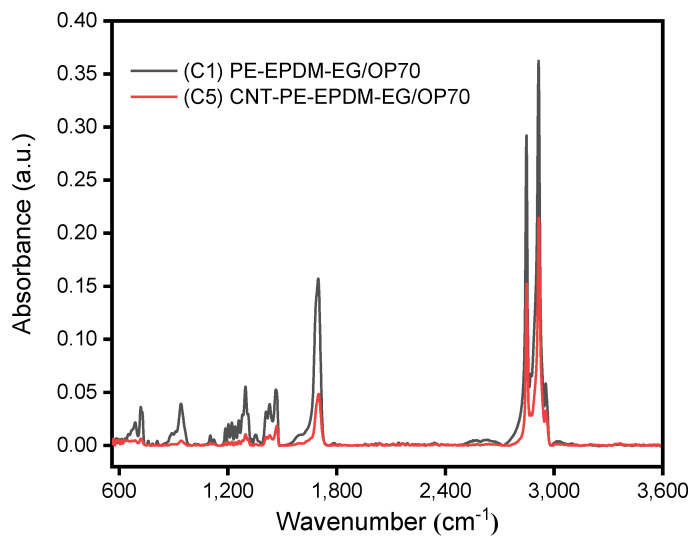
ATR FT-IR spectra of PE-EPDM/EG-based (C1) and CNT/PE-EPDM/EG-based (C5) phase change composites.

**Figure 6 polymers-14-02285-f006:**
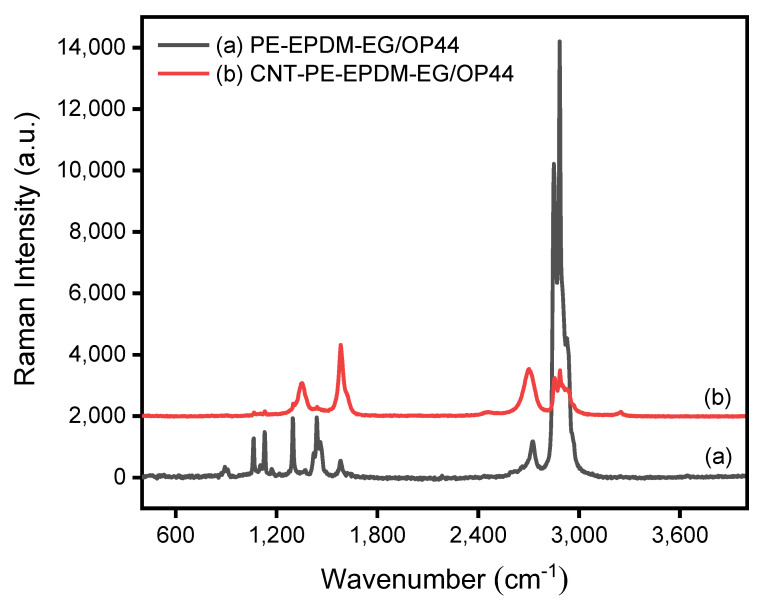
Raman spectra of PE-EPDM/EG-based (C1) and CNT/PE-EPDM/EG-based (C5) phase change composites.

**Figure 7 polymers-14-02285-f007:**
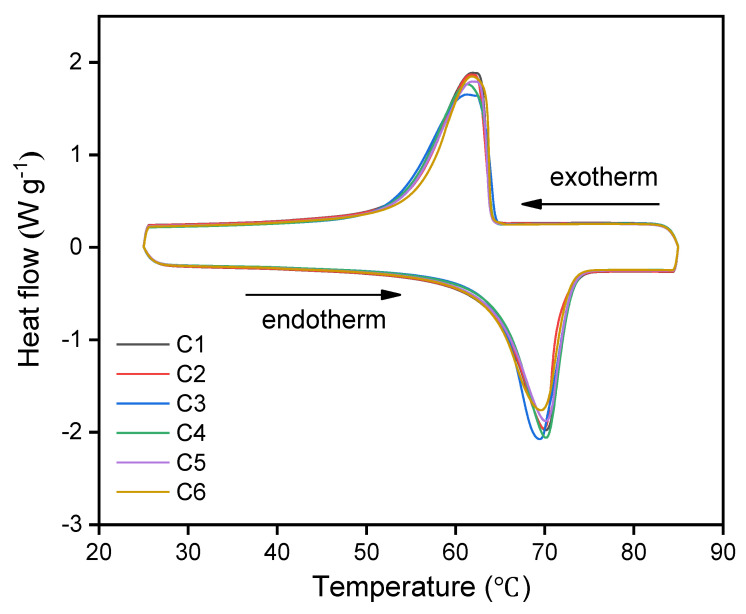
DSC curves of the as-prepared samples.

**Figure 8 polymers-14-02285-f008:**
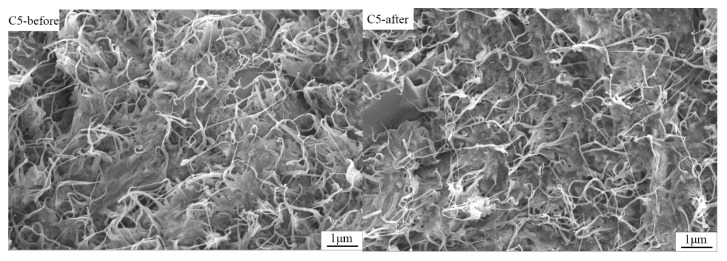
SEM images of CNT-PE-EPDM/EG-based PCM sample C5 before heating-cooling test and after 200 cycles.

**Figure 9 polymers-14-02285-f009:**
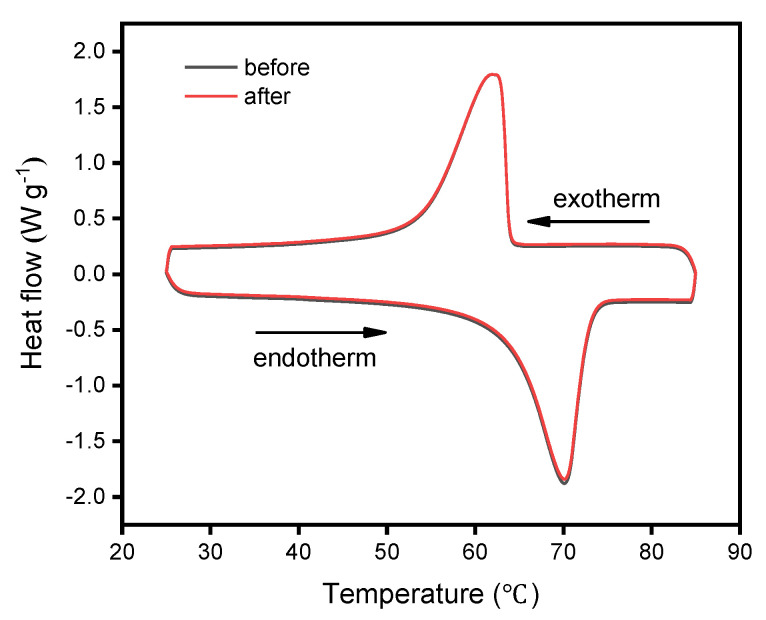
Melting−freezing properties of CNT-PE-EPDM/EG-based PCM sample C5 before and after 200 cycles.

**Figure 10 polymers-14-02285-f010:**
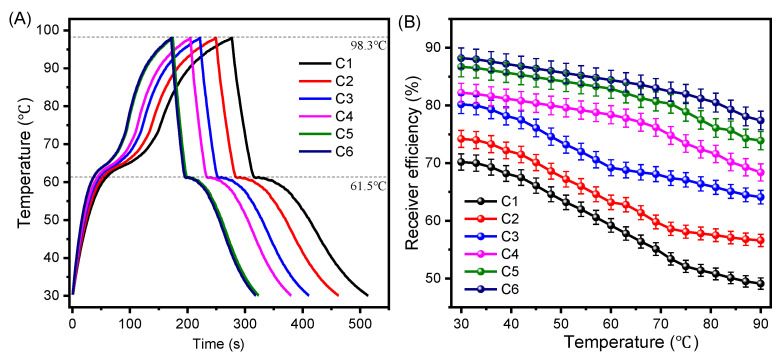
Photo-thermal performance (**A**) and receiver efficiency (**B**) the as-prepared CNT-PE-EPDM/EG-based PCM samples.

**Table 1 polymers-14-02285-t001:** Mass (g) of the as-prepared thermal conductivity enhanced phase change composite material samples.

Samples	PE (g)	EPDM (g)	OP70/EG ^a^ (g)	CNT (g)	Total (g)
C1	30	5	70	0	105
C2	30	5	70	1	106
C3	30	5	70	2	107
C4	30	5	70	4	109
C5	30	5	70	6	111
C6	30	5	70	8	113

^a^ mass ratio of OP70/EG (OP70:EG = 9:1).

**Table 2 polymers-14-02285-t002:** Mass percentage of the as-prepared thermal conductivity enhanced phase change composite material samples (%).

Samples	PE	EPDM	OP70	EG	CNT	Total
C1	28.57	4.76	60.00	6.67	0.00	100.00
C2	28.30	4.72	59.44	6.60	0.94	100.00
C3	28.04	4.67	58.88	6.54	1.87	100.00
C4	27.52	4.59	57.80	6.42	3.67	100.00
C5	27.03	4.50	56.76	6.31	5.40	100.00
C6	26.55	4.43	55.75	6.19	7.08	100.00

**Table 3 polymers-14-02285-t003:** Melting-freezing properties for CNT-PE-EPDM/EG base composites.

Samples	T_m_/°C	T_m, max_/°C	△H_m_/J g^−1^	T_f_/°C	T_f, max_/°C	△H_f_/J g^−1^
C1	63.8	70.2	138.9	63.9	62.1	138.4
C2	63.9	69.9	137.6	64.2	61.8	137.4
C3	64.1	69.4	135.7	64.0	61.6	135.5
C4	64.6	70.0	133.2	64.1	61.3	133.9
C5	64.5	70.2	130.3	64.2	62.1	130.5
C6	64.4	69.6	129.1	64.3	61.9	129.4

## Data Availability

Not applicable.

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
