# Peer review of "Construction of Three-Dimensional Network Structure in Polyethylene-EPDM-Based Phase Change Materials by Carbon Nanotube with Enhanced Thermal Conductivity, Mechanical Property and Photo-Thermal Conversion Performance"

_polymers, 2022, doi:10.3390/polym14112285_

Round 1
Reviewer 1 Report
Title: “Construction of three-dimensional network structure in poly-ethylene-EPDM based phase change materials by carbon nano-tube with enhanced thermal conductivity, mechanical property 4
and photo-thermal conversion performance“
In this work, the authors constructed a structure in polyethylene (PE) and ethylene-propylene-diene monomer (EPDM) based phase change composites by mixing with carbon nanotube (CNT). In particular, two-dimensional flake expanded graphite in PE-EPDM based phase change materials and one-dimensional CN were cross-linked with each other to build dense three-dimensional thermal pathways. In this work the authors claim that CNT (5.4%wt)-PE-EPDM phase change composite shows excellent thermal conductivity (3.11 W m-1 K-1) and mechanical property with tensile and bending strength of 10.19 and 21.48 MPa. In addition, the melting and freezing temperature of the optimized phase change composite are measured to be 63.2 2and 63.9 °C and the melting and freezing latent enthalpy are measured to be 138.7 and 135.1 g-1.
General comment: Although the aim of this work could be interesting, this work should be revised to improve its quality and impact. In particular, the value of this work is not clear with respect to the current state of the art. It is not clear, whether the proposed three-dimensional network structure is totally novel and whether the described properties are important. Finally, a “Results” and a “Discussion” sections are strongly suggested. Indeed, from a side the authors should present in details the main results of this research, from the other they should clearly compare their main results to the current state of the art, discussing the value of their work with reference to the state of the art in the field.
Some detailed comments:
Lines:” 2.2. Preparations of CNT-Polyethylene-EPDM based phase change materials 103
The OP70/EG sample was prepared from EG absorbing OP70 with mass ratio of 1:9. 104
Then, 70 parts by mass of the obtained OP70/EG, 5 parts by mass of ethylene propylene 105
rubber and 30 parts by mass of high-density polyethylene were mixed through open mill- 106
ing procedure (XH-401, Dongguan Xihua Co., Ltd., PR. China) [24,25]. To boost the ther- 107
mal conductivity and the mechanical property, 1, 2, 4, 6, 8 parts by mass of CNT were 108
added respectively, following sequential blending. The mass ratio and mass percentage 109
of the as-prepared samples are present in Table 1 and Table 2. The as-prepared phase 110
change composites CNT-PE-EPDM-EG/OP70 were fabricated into blocks by hot-press 111
(XH-406B, Dongguan Xihua Co., Ltd.). 112”
“Table 1 Mass ratio of the as-prepared thermal conductivity enhanced phase change composite material samples”
“Table 2 Mass percentage of the as-prepared thermal conductivity enhanced phase change composite material samples (%) “
*) Tables 1 and 2 are not totally clear. Please improve their presentation together with their captions.
Section: “3. Results and discussion”
*) This section should be split in “Results”, where the authors present in a detailed way the main results and a “Discussion” section, where they compare the value of the work with respect to the current state of the art.
Figure 3 Mechanism of thermal conductivity enhancement for CNT-PE-EPDM/EG based PCMs
*) This figure is not clear. Please provide a better caption.
Lines: “3.3. Mechanical property of the as-prepared CNT/PE-EPDM/EG based PCMs 190
The tensile and bending strength of the as-prepared CNT/PE-EPDM/EG based 191
PCMs are measured. As present in Figure 4A, the tensile strength of the PCMs increases 192
from 8.39 MPa (sample C1) to 10.19 MPa (sample C5) in the case that the mass fraction of 193
CNT increases from 0.9 to 5.4%. However, the tensile strength of sample C6 decreases to 194
9.11 MPa while the mass fraction of CNT increased to 7.1%. Similarly, as shown in Figure 195
4B, the bending strength of the PCMs increases from 14.67 MPa (sample C1) to 21.48 MPa 196
(sample C5), but the bending strength of sample C6 decreases to 13.65 MPa while the mass 197
fraction of CNT increased to 7.1%. This indicated that a small amount of CNT could pro- 198
mote the structural of CNT/PE-EPDM/EG based PCMs compact and enhance the tensile 199
and bending strength. However, when the addition amount exceeds the critical value, the 200
continuous structure of polyethylene-EPDM/EG based PCMs is destroyed, causing a 201
sharp drop in mechanical strength. In virtue of the good mechanical properties, the 202
CNT/PE-EPDM/EG based PCMs can be served as heat storage medium for solar thermal 203
energy storage to achieve cooling or heating effectiveness.”
Figure 4 Tensile strength (A) and bending strength (B) of CNT-PE-EPDM/EG based PCMs
*) This paragraph should be reworked in order to better explain all the mechanical tests used to characterize the CNT/PE-EPDM/EG based PCMs. In particular, a detailed description of the mechanical test used to obtain all results in Figure4, together with clear and detailed plots should be inserted within the main text.
Lines: “3.5. Melting and freezing behavior of CNT/PE-EPDM/EG phase change material
The DSC curves and melting-freezing properties of the as-prepared PE-EPDM/EG 243
based PCM samples are compared in Figure 7 and Table 3. The melting and freezing tem- 244
peratures of PE-EPDM/EG based PCM are 63.8 and 63.9 ℃, corresponding to the latent 245
enthalpies of 138.9 and 138.4 J g-1, respectively. With the increasing amount of CNT from 246
0.9 to 7.1wt%, the melting latent enthalpy of CNT/PE-EPDM/EG based phase change ma- 247
terials decreases from 137.6 to 129.1 J g-1. The enthalpy of CNT/PE-EPDM/EG based phase 248
change material is reduced at a minor value of about 7.9 J g-1 and its thermal conductivity 249
is improved from 1.99 to 3.11 W/(mK) with a small amount of CNT addition (C1 to C5). 250
It is worth noting that highly similar curves were collected for all the PCMs samples. Low 251
heat transfer lag in DSC test was observed due to their high thermal conductivities. 252
Figure 7 DSC curves of the as-prepared samples 25”
*) These lines are not clear. Please explain better and insert a more detailed caption.
Lines: “3.7 Photo-thermal conversion performance of CNT/PE-EPDM/EG phase change materials 275
The photo-thermal conversion performance and receiver efficiency of the as-pre- 276
pared CNT-PE-EPDM/EG composite PCMs are shown in Figure 10. It can be found that 277
temperature plateau between 65~70 ℃ appears in the both heating and cooling process, 278
showing the typical characteristic of the temperature-time curve of phase change materi- 279
als. The time for sample C1 to increase from room temperature to 97℃ is 274s, while the 280
time for sample C2 to C6 are 237, 211, 195, 166 and 163s, respectively. From sample C1 to 281
C6, the temperature rise time is reduced by 40%. These results indicate that the tempera- 282
ture growth rate is positively correlated with the increasing thermal conductivity. The 283
receiver efficiency is calculated according to Equation (1). 284
= 100% T
Q s
G At
(1) 285
Where the η is the receiver efficiency, Gs is the irradiance of solar radiation, A is the 286
irradiation area, t is the irradiation time. The calculated receiver efficiencies are shown in 287
Figure 10(B). It is shown that the receiver efficiency of all samples presents a downward 288
trend with the raise of temperature. It is mainly because of the higher of temperature, the 289
more heat flows into the atmosphere. The receiver efficiency of sample C1 decreases from 290
70% to 50% as the temperature raised from 30 to 90℃. The receiver efficiency from sample 291
C1 to C6 increases in turn under the same receiver temperature. The receiver efficiency of 292
sample C5 decreases from 87% to 74% at as the temperature raised from 30 to 90℃. The 293
receiver efficiency of sample C6 reaches 88% at 30℃ and over 80% at 90℃. These results 294
demonstrate that high thermal conductivity in CNT/PE-EPDM/EG phase change material 295
*) Please improve this paragraph. All formulas should be moved and described within the “Materials and Methods” section. Please correct.
Figure 10 Photo-thermal performance (A) and receiver efficiency (B) the as-prepared CNT-PE-EPDM/EG 299
based PCM samples
*) These figures should be made more clear through better captions. Please rework.
Lines: “4. Conclusions 301
In this work, we constructed a three-dimensional network structure in polyethylene 302
and ethylene-propylene-diene monomer-based phase change materials by adding carbon 303
nanotube. The as-prepared CNT-PE-EPDM/EG based phase change material with high 304
thermal conductivity and good mechanical property exhibits high latent heat and good 305
photo-thermal performance for effective utilization of solar thermal energy. The opti- 306
mized mass fraction of CNT in CNT-PE-EPDM/EG phase change composite is 5.4%. The 307
optimized CNT-PE-EPDM/EG based phase change material has a thermal conductivity of 308
3.11 W m-1 K-1 with the constructed dense three-dimensional thermal pathways by CNT 309
addition, and has good mechanical property with tensile and bending strength of 10.19 310
and 21.48 MPa. The melting and freezing latent enthalpy of the PCM are estimated to be 311
138.7 and 135.1 J g-1, respectively. The FT-IR and Raman results indicate the addition of 312
CNT can effectively improve the encapsulation of paraffin for preventing the leakage of 313
paraffin in CNT/PE-EPDM/EG based PCM. The CNT/PE-EPDM/EG based PCM shows 314
great photo-thermal performance with 60.6% of heating storage time and 1.48 time of re- 315
ceiver efficiency than PE-EPDM/EG based PCM. The CNT/PE-EPDM/EG based PCM is 316
expected to be one of advanced materials for photo-thermal energy storage in solar ther- 317
mal utilization. 31”
*) This section should be enlarged and the value of the work better underlined. The lines “The CNT/PE-EPDM/EG based PCM is 316
expected to be one of advanced materials for photo-thermal energy storage in solar ther- 317
mal utilization. 31” should be better explained to the interested readers.
Reviewer 2 Report
The paper presents the construction of three-dimensional network structure in poly-ethylene-EPDM based phase change materials by carbon nanotube with enhanced thermal conductivity, mechanical property and photo-thermal conversion performance. Over an entire manuscript, objectives, experimental results and analysis of the paper are well described without any default. The paper can be accepted for publication in present form.
Reviewer 3 Report
The manuscript under consideration is devoted to the development of new "smart" polymer composite material with an increased heat capacity, capable of heating and storing heat by absorbing solar energy with subsequent heat output when the ambient temperature decreases. The material belongs to the so-called "phase transition material". It contains the following components: paraffin with a melting point of 65 C, expanded nano graphite, high-density polyethylene and multi-walled nanotubes. The key characteristics of the material are high heat capacity (135 J/g) and high thermal conductivity (up to 3.1 W/ K*m(2). The first parameter is achieved due to the enthalpy of the reversible melting process – the crystallization of paraffin, which is part of the material. The high thermal conductivity of the material is achieved due to the successful combination of expanded nanoparticles of expanded graphite and carbon nanotubes, which form a continuous percolation structure. The dependence of thermal conductivity on the content of nanotubes has been studied. The obtained samples showed good reproducibility in numerous heating-cooling cycles. Compared with known composite materials of this type based on porous polymers and paraffin, the materials developed in this work have good mechanical properties associated with the presence of a high-molecular component in the composite material. In general, the manuscript makes a good impression, the authors managed to significantly improve the thermal conductivity of the polymer composite material in comparison with others by introducing nanotubes and at the same time achieve high mechanical properties.
The following critical comments can be made on the work:
1) Morphological structure of composite material is not clear from manuscript. It remains unclear how it is constructed: does the 3D structure mean a three-dimensional percolation structure formed by expanded graphite particles and nanotubes, or does the material itself have an independent structural and mechanical framework? What phase is the polymer in? The authors point out that they have detailed structural morphological data in their previous article, but, from our point of view, in this article it is necessary to give at least general information about phase morphology.
2) It is unclear what role the high polymer plays? What phase is it in? what is the compatibility in a sentence “…effect of PE and EPDM/EG on the compatibility of EG/paraffin..” (page 2, line 70)?
3) It is not clear what kind of “inter-reaction” the authors have in mind (page 2, line 72)?
4) A mistake was made in the phrase: “bending strength”(page 2, line 61) should be replaced with “bending strength”.
5) What do mean sentences “…band of bending vibration of O-H, which corresponds to carboxyl of paraffin (p.6, line 213)...” and “…The bands at 1700 cm(-1) are contributed to the C=O stretching vibration modes of carboxyl in paraffin…”(p.6, line 216). Some doubts appear concerned is it paraffin but not stearin acid?
6) The text should be carefully edited (page 2, lines 45-49).
After making the necessary additions, the manuscript can be recommended for publication in the journal Polymers.
Round 2
Reviewer 1 Report
Title: “Construction of three-dimensional network structure in poly-ethylene-EPDM based phase change materials by carbon nano-tube with enhanced thermal conductivity, mechanical property and photo-thermal conversion performance“
In this work, the authors constructed a structure in polyethylene (PE) and ethylene-propylene-diene monomer (EPDM) based phase change composites by mixing with carbon nanotube (CNT). In particular, two-dimensional flake expanded graphite in PE-EPDM based phase change materials and one-dimensional CN were cross-linked with each other to build dense three-dimensional thermal pathways. In this work the authors claim that CNT (5.4%wt)-PE-EPDM phase change composite shows excellent thermal conductivity (3.11 W m-1 K-1) and mechanical property with tensile and bending strength of 10.19 and 21.48 MPa. In addition, the melting and freezing temperature of the optimized phase change composite are measured to be 63.2 2and 63.9 °C and the melting and freezing latent enthalpy are measured to be 138.7 and 135.1 g-1.
General comment: Although the authors revised this version of the main text, some further issues should be corrected. In addition, the quality of the main text is somewhere sub optimal. Please correct.
Some detailed comments:
"Table 2"
* )The sums of percentages should be controlled, since it is greather that 100% or smaller...
C1 28.6 4.76 60 6.67 0 100.03
C2 28.3 4.72 59.4 6.6 0.9 99.92
C3 28 4.67 58.9 6.54 1.9 100.01
C4 27.5 4.59 57.8 6.42 3.7 100.01
C5 27 4.5 56.8 6.31 5.4 100.01
C6 26.5 4.42 55.8 6.19 7.1 100.01
lines: "The 212
CNT/PE-EPDM/EG based PCMs can be served as heat storage medium for solar thermal energy storage to achieve cooling or heating effectiveness"
*) This claim should be better explained since the caractheristic times for cooling or heating should be explicitly quantified.
lines: "Figure 8 SEM images of CNT-PE-EPDM/EG based PCM sample C5 before and after 200 cycles"
*) This caption is not clear. Please explicit the meaning of "before" ... is it the 199 cycle or before all cycles ?
*)Plese improve the quality of text in order to allow the interested readers to understand the main flow of the work.
Reviewer 4 Report
My comments are satisfactorily addressed in the revised version of the manuscript. However there are still some typos, also in sentences added to the text for the revision. I recommend their corrections.
